# Quantification and Distribution of Thiols in Fermented Grains of Sauce-Aroma Baijiu Production Process

**DOI:** 10.3390/foods12142658

**Published:** 2023-07-10

**Authors:** Danhua Xiang, Peiqi Li, Rong Gong, Yanbin Sun, Xiangmei Chen, Heli Wei, Yan Xu

**Affiliations:** 1Laboratory of Brewing Microbiology and Applied Enzymology, School of Biotechnology, Jiangnan University, Wuxi 214100, China; 6200201093@stu.jiangnan.edu.cn (D.X.); 18101536750@163.com (P.L.); 2Guizhou Jinsha Liquor Wine Cellar Co., Ltd., Bijie 551800, China; drgongrong@163.com (R.G.); sunyanb163@163.com (Y.S.); cxm7128@163.com (X.C.); 15085333040@139.com (H.W.); 3Key Laboratory of Baijiu Supervision Technology for State Market Regulation, Chengdu 610097, China

**Keywords:** volatile thiol compounds, fermented grains, UPLC-MS/MS, QuEChERS, derivatization, sauce-aroma baijiu

## Abstract

Five volatile thiol compounds (methanethiol, ethanethiol, 2-mercapto-1-ethanol, 2-furfurylthiol, and 2-methyl-3-furanethiol) in fermented grains of sauce-aroma baijiu were determined using ultra-performance liquid chromatography–tandem mass spectrometry (UPLC-MS/MS). The samples were pre-treated using a modified QuEChERS method. 4,4′-Dithiodipyridine (DTDP) derivatization reaction improved the detectability and stability of volatile thiol compounds. From the end of the first round to the end of the seventh round of fermentation and different fermentation states from the fifth round of fermented grains of the sauce-aroma baijiu production process were analyzed. The results showed that the concentrations of methanethiol (67.64–205.37 μg/kg), ethanethiol (1.22–1.76 μg/kg), 2-furfurylthiol (0.51–3.03 μg/kg), and 2-methyl-3-furanthiol (1.70–12.74 μg/kg) were increased with the number of fermentation rounds. Methanethiol, 2-furfurylthiol, and 2-methyl-3-furanthiol increased during fermentation and distillation in the fifth round. Fermentation and distillation were important stages for their widespread production. After distillation, there were still a large number of volatile thiol compounds in the fermented grains. The thermal reaction was of great significance in the formation of these thiols.

## 1. Introduction

Volatile thiol compounds in alcoholic beverages (wine, baijiu, beer, etc.) are important contributors to sensory aroma. These thiols with low thresholds exhibit powerful and characteristic odors [1,2] that can contribute to either pleasant or unpleasant aromas in alcoholic beverages, depending on their nature and concentration [3,4,5]. Such compounds include 2-methyl-3-furanethiol (odor thresholds: 0.0048 μg/L in 46% ethanol-water [6], roasted meat, fried), 2-furfurylthiol (0.1 μg/L, coffee, roasted sesame seeds), ethanethiol (0.8 μg/L, onion, rubber), methanethiol (2.2 μg/L, burned rubber, gasoline), etc. In the last decade, volatile thiols, which are important for sensory qualities, have gained widespread interest in baijiu [7,8,9]. The concentrations of these volatile thiol compounds are well above their low odor thresholds in baijiu [10]. However, an excessive concentration of thiols results in an abnormal smell that gives sauce-aroma baijiu its pickle-like off-odor [11,12]. This abnormal smell eventually degrades the quality of baijiu. Balancing the concentrations of these potentially off-odor compounds in alcoholic beverages is currently a major challenge. Therefore, determining the concentration of thiols and gaining insight into their production during the fermentation process will be beneficial for achieving a balance of thiols in alcoholic beverages.

Low concentrations, poor stability, and the complexity of the matrix make the qualitative and quantitative analysis of volatile thiol compounds with various odor flavors difficult [13]. Thus, the methods for detecting thiols must be reliable and accurate. Volatile thiol compounds in alcoholic beverages have been analyzed using various techniques. Gas chromatography analysis has been widely utilized [1,14,15]. Owing to their low content, the detection of volatile sulfides generally necessitates the use of specific detectors. Pulsed flame photometric detection (PFPD) [16] and sulfur chemiluminescence detection (SCD) [17] can effectively exclude interference from other compounds and improve the sensitivity and accuracy of sulfide detection. When using common mass spectrometry detectors, additional derivatization is required to enhance the stability and detectability of thiol detection [18,19]. However, gas chromatography cannot handle the high-temperature oxidation of volatile thiol compounds, resulting in erroneous quantification.

The detection method of volatile thiols in liquid liquor has been solved [20], but the detection method of volatile thiols in a solid matrix needs to be developed urgently. The raw materials for the production of sauce-aroma baijiu are solid fermented grains [21], which make the extraction of their flavor compounds very difficult. This is especially true for the extraction of volatile sulfides. The quick, easy, cheap, effective, rugged, and safe (QuEChERS) method can efficiently handle the challenge of extracting active chemicals from solid matrices. Thiols in wine have been pre-treated using the QuEChERS method [22]. There has not been any investigation into whether QuEChERS can be a technique for detecting the pre-treatment of thiols in the alcoholic beverages production process, particularly for the intricate baijiu production process. Ultra-performance liquid chromatography–tandem mass spectrometry (UPLC-MS/MS) combines the technical advantages of UPLC and MS/MS and is useful for detecting compounds with high boiling points, strong polarities, poor thermal stabilities, and low content. Cold injection of the liquid phase combined with derivatization can resolve the problems of thiol detectability and high-temperature instability.

This study aimed to develop a method for determining volatile thiol compounds in fermented grains. The improved QuEChERS pre-treatment method, in conjunction with the 4,4′-dithiodipyridine (DTDP) derivatization reaction, was employed to measure the content variation law of volatile thiol compounds in fermented grains over seven rounds of heap fermentation and distillation. This provided new insight into the analysis and determination of control points for baijiu production to improve its quality.

## 2. Materials and Methods

### 2.1. Standards, Chemicals, and Materials

The thiols investigated were methanethiol, ethanethiol, 2-mercapto-1-ethanol, 2-furfurylthiol, 2-methyl-3-furanethiol, and 2-phenylethanethiol. Commercially available high-purity analytes (96%) were obtained from Sigma-Aldrich (Shanghai, China). DTDP (97%) was purchased from Sigma-Aldrich (Shanghai, China). Ethylenediaminetetraacetic acid disodium salt (EDTA-Na_2_), acetaldehyde (99%), and formic acid (99%) were purchased from J&K Chemical Corp., (Beijing, China). Acetonitrile (ACN, LC-MS, HPLC grade), methanol (MeOH, HPLC grade), ethanol (EtOH > 99.8%), ethyl acetate (EAC, HPLC grade), dichloromethane (DCM, HPLC grade), Anhydrous magnesium sulfate (MgSO_4_), sodium chloride (NaCl), sodium citrate tribasic dihydrate (C_6_H_5_Na_3_O_7_·2 H_2_O, 99.0%), and sodium citrate dibasic sesquihydrate (C_6_H_6_Na_2_O_7_·1.5 H_2_O, ≥ 99.0%) were supplied by Sinopharm Chemical Reagent Co., Ltd. (Shanghai, China). Aminopropyl-modified silica (NH_2_), graphitized carbon black (GCB), and octadecyl-modified silica (C_18_) were purchased from the ANPEL Laboratory Technologies (Shanghai, China). Ultra-pure water (18 MΩ cm) was obtained using a Milli-Q water purification system (Millipore, Billerica, MA, USA).

### 2.2. Fermented Grains of Baijiu

The production of sauce-aroma baijiu was carried out by feeding only two batches of fresh grains. After the first round of distillation, the grains after distillation were spread on the ground to cool and mixed with daqu powder, and distillates were collected from the first round of distillation, ready for the second round of heap fermentation and alcoholic fermentation. Seven iterations of the fermentation processes were carried out, from heap fermentation to alcoholic fermentation. From the end of the first round to the end of the seventh round of fermentation, fermented grains of baijiu from Chinese sauce-aroma baijiu–producing regions (Maotai Town, Guizhou Province) were collected. Additionally, initial grains, heaped grains, fermented grains, distilled grains, and samples from the fifth round of fermented grains of baijiu were collected from another batch. Fermented and distilled grains were selected from the upper, middle, and bottom layers. All samples were collected from 2020 to 2021 and stored at −20 °C for further analysis. Triplicates for each grain sample were used.

### 2.3. Derivatizing Reagent

The reagent preparation procedure was as follows: 220 mg of DTDP were sonicated in a mixture of 20 mL of water and 100 μL of concentrated HCl (37% w/w). To the dissolved DTDP solution, Milli-Q water was added to a final volume of 100 mL. Aliquots were then stored at –20 °C until needed [23].

### 2.4. Optimization of the QuEChERS Method

Acetonitrile is a frequent extractant used in QuEChERS [22]. The overall effectiveness of extraction was set using various extractants [24]. To ensure that the molecules of interest were correctly extracted from the matrix, the performance of various commonly used solvents was assessed. Acetonitrile, methanol, dichloromethane, ethanol, and ethyl acetate were commonly utilized in reported extraction processes. The aforementioned five typical solvents were chosen, but ethyl acetate cannot be directly injected. When ethyl acetate was used as an extractant, it was nitrogen-blown dried before being redissolved in 400 μL of methanol.

The d-SPE phase was the second stage of the QuEChERS method. Primary secondary amine (PSA), C_18_, and GCB are the most widely reported adsorbents. NH_2_ and PSA have similar functions. They are used to clear lipids, organic acids, carbohydrates, fatty acids, and certain pigments. These sorbents are typically employed alone or in combination, and certain purification procedures can be avoided. To optimize the three purifying agents (NH_2_, GCB, and C_18_) as samples for optimization, fermented grains of baijiu from the fifth fermentation stage were chosen.

### 2.5. Preparation of Sample for Analysis

Figure 1 shows the entire process, from the pre-treatment to the injection stage. The sample was made up of 10 g fermented grains of sauce-aroma baijiu and 5 mL of water in a 50 mL centrifuge tube. It was then placed in an ice-water bath for 5 min using an ultrasonic cleaner. An ACN solution (10 mL) (stored at −20 °C) containing 10 µL 2-phenylethanethiol (internal standard) at a concentration of 58.7 mg/L was added to the tube and mixed by vortex shaking. The samples were then prepared using the QuEChERS method. MgSO_4_ (4 g), NaCl (1 g), sodium citrate tribasic dihydrate (1 g), and sodium citrate dibasic sesquihydrate (0.5 g) were added to the tube, which was immediately vortexed for 2 min, hand-shaken for 30 s, chilled in an ice-water bath, and centrifuged for 5 min at 4500 rpm and 4 °C. In an ice-water bath, the supernatant (5 mL) was transferred to a new 15 mL tube. Subsequently, EDTA-Na_2_ (5 mg), 50% acetaldehyde (20 μL), and freshly thawed DTDP reagent (10 mM, 400 μL) were added to the tube. After 30 min, the tubes were then dried in a stream of nitrogen until the sample volume reached 400 μL [6]. The final solution was filtered through a 0.22 μm PTFE membrane before the UPLC-MS/MS analysis.

### 2.6. UPLC-MS/MS Analysis

UPLC-MS/MS analysis was performed using an ACQUITY UPLC system (binary solvent manager; Waters Corp., Milford, CT, USA). Chromatographic separation was performed on a C18 column (100 × 2.1 mm, 1.7 μm; Waters BEH C18). The column temperature was maintained at 40 °C. The liquid chromatography eluent consisted of a gradient of water (mobile phase A) and ACN (mobile phase B), both of which contained 0.1% formic acid. The program was performed as follows: 0–13 min, 15%–22% B; 13–14 min, 22–30% B; 14–18 min, 30–35% B; 18–18.5 min, 35–100% B; 18.5–21.5 min, 100% B; and 21.5–22 min, 100%–15% B. The flow rate was 0.3 mL/min, and the injection volume was 10 µL. A triple-quadrupole mass spectrometer (Xevo TQ-S; Waters Corp.) was used with an electrospray ionization source. The scanning mode was set to the positive ionization mode. The following mass spectrometry detection conditions were chosen: source temperature, 150 °C; desolvation temperature, 500 °C; and capillary voltage, 3 kV. The multiple reaction monitoring (MRM) mode was used for targeted compound analysis. The derivatization products were optimized via direct injection into LC-MS to determine the MRM conditions, and the optimized parameters of volatile thiols in baijiu were characterized by Yan Yan [6]. The derivatization reaction was a chemical reaction with a chemical dose ratio of 1:1, and the DTDP measured in the sample guaranteed excess derivatization. The retention times determined by reaction with the standard in the sample were the same. Thus, the content of the derivatization product might represent that of the original compound. The content was calculated based on the internal standard, and the peak area of the compound was determined in the MRM mode.

The retention times, parent ions, and daughter ions of the five volatile thiol compounds are displayed in Figure 2, and detailed information on the MRM method for the thiol compounds is displayed in Appendix A. The cone voltage and collision energy of these compounds were 23 and 19 for methanethiol, 23 and 19 for ethanethiol, 21 and 15 for 2-mercapto-1-ethanol, 21 and 19 for 2-furfurylthiol, and 23 and 25 for 2-methyl-3-furanethiol, respectively. The retention time, parent ion, product ion, cone voltage, and collision energy of 2-phenylethanethiol (internal standard) were 15.39 min, 248.5, 143.5, 23, and 20, respectively.

### 2.7. Statistical Analyses

The UPLC-MS/MS data were processed using Mass Lynx V4.1. Statistical analysis was performed using SPSS software (Version 26.0; Chicago, IL, USA) and Microsoft Excel 2019. The drawings were completed using Origin Pro 2022b.

## 3. Results and Discussion

### 3.1. Optimization of the Extraction Solvent

The effect of extracting thiols from the fermented grains with five commonly used organic reagents was optimized. Methanethiol produced the best response, and the remaining four thiols were pre-treated and then detected in other organic reagents and could not even be detected or provided a bad response. Thus, Figure 3 demonstrates the effect of methanethiol on the extraction of various organic reagents. The abundance of methanethiol extracted from the fermented grains using acetonitrile was approximately 10^6^, while the detection after extraction using the rest of the organic reagents was approximately only 10^4^ to 10^5^. Comparing the results of the other four thiols after extraction using different organic reagents, the results were the same as those using methanethiol.

The response of the extracted thiol compounds was the highest when acetonitrile was employed. Other extraction solvents resulted in poor extraction results, low abundance, and numerous heterogeneous peaks. This is because acetonitrile is a polar solvent with low lipophilicity, which allows good recovery of polar chemicals while reducing fat, wax, and pigment extraction compared to other solvents. Acetonitrile was selected as the final extraction solvent.

### 3.2. Selection of Clean-Up QuEChERS Extracts

To optimize the three purifying agents (NH_2_, GCB, and C_18_) as samples for optimization, fermented grains of baijiu from the fifth fermentation stage were chosen. Mixes of GCB (20 mg), C_18_ (120 mg), and NH_2_ (120 mg) were chosen along with a blank control group [25] (Figure 4). Combination and separate adsorbents were used to extract volatile thiols from the solid materials. The extraction effects of five volatile thiols were considered together. The use of C_18_ alone resulted in very poor detection of methanethiol, 2-furfuranethiol, and 2-mercapto-1-ethanol, and GCB alone resulted in very poor detection of 2-mercapto-1-ethanol. Furthermore, the use of NH_2_ alone resulted in poor detection of 2-mercapto-1-ethanol, ethanethiol, and 2-methyl-3-furanethiol, and the combination of purifying agents resulted in very poor detection of 2-methyl-3-furanethiol.

It was discovered that skipping the purification phase was more effective than using adsorbents. This might be because the DTDP derivatization reaction worked well at pH ≥ 3.4 and reacted rapidly and completely [20]. However, the purifiers, particularly NH_2_, were alkaline in solution, and after adding the purification step, the pH of the extract became alkaline, causing the derivatives to become unstable and deteriorate in an alkaline environment. As a result, the ultimate option was to skip the purification step.

### 3.3. Validation of the Method

Thiol compound standard solutions of different concentrations were obtained by diluting the stock solutions in Milli-Q water. The different concentration levels were determined using UPLC-MS/MS under the pre-treatment settings described above. Calibration values were calculated by comparing the peak area ratio of each thiol component against various internal standard concentrations (2-phenylethanethiol). Good coefficients of determination (R^2^: 0.9900–0.9935) were observed between the concentration and the matching peak area of each thiol. The limits of detection (LODs) and limits of quantitation (LOQs) of the diluted thiol compounds were determined using continuous analysis. The LODs and LOQs were calculated when the signal/noise ratio (S/N) reached 3 and 10, respectively. Detailed information is presented in Table 1.

The accuracy of the method was assessed based on the recovery rate and evaluated by adding three spiked levels (high, medium, and low) to test the target compound in the fermented grains of baijiu. Samples from the fifth round of fermented grains were chosen to validate the research method. In repeated experiments, the samples were pre-treated using the previously reported QuEChERS method, as well as a derivatization reaction; the final solution was injected into the LC-MS/MS system for analysis. The recoveries of the five volatile thiol compounds in fermented grains of baijiu were 71.72–104.72%, and the intra- and inter-day relative standard deviations (RSDs) were 0.63–7.72% and 1.96–9.44%, respectively. The recoveries were within the acceptable range of 70–110% in terms of accuracy. Additionally, accuracy was determined, with RSD values less than 15%, supporting the accuracy of the method. The results are presented in Table 2.

### 3.4. Measurement of Fermented Grains of Baijiu

For this study, the QuEChERS binding derivatization pre-treatment approach was used for fermented grains. Five volatile thiol compounds were identified in the fermented grains of baijiu. Figure 5 displays the quantitative results of the five thiol compounds from the first to seventh rounds of fermented grains. The standard curve uses the concentration of the compound as the abscissa by calculating the concentration of the derivatized product to represent the actual content of the thiol compound in the sample. The quantitative results showed that the concentrations of thiols from the first to seventh rounds of fermented grains were methanethiol (67.64–205.37 μg/kg), ethanethiol (1.22–1.76 μg/kg), 2-mercapto-1-ethanol (1.12–1.85 μg/kg), 2-furfurylthiol (0.51–3.03 μg/kg), and 2-methyl-3-furanthiol (1.70–12.74 μg/kg) (Appendix A). The concentrations of four thiols (methanethiol, ethanethiol, 2-furfurylthiol, and 2-methyl-3-furanthiol) increased with the number of fermentation rounds. This implies that, between the fermentation rounds, volatile thiols may accumulate in the preceding round, resulting in an increase in the content of the next round. It was reported that the concentrations and odor thresholds in 46% ethanol-water of five thiols of 229–513 and 2.2 μg/L for methanethiol, 6.7–32.1 and 0.8 μg/L for ethanethiol, 0.03–0.08 and 130 μg/L for 2-mercapto-1-ethanol, 11.2–37.8 and 0.1 μg/L for 2-furfurylthiol, and 1.0–2.5 and 0.0048 μg/L 2-methyl-3-furanthiol, respectively, were determined in the sauce-aroma baijiu [6]. The concentrations of these volatile thiol compounds were closely correlated with their counterparts in the raw materials during baijiu fermentation.

The concentration of volatile thiol compounds in sauce-aroma baijiu increased as the number of rounds increased, but it was not clear at which stage in the production of a single round the most volatile thiol compound accumulated. Therefore, the content of volatile thiol compounds in the fifth round of fermented grains (initial, heaped, fermented, and distilled grains) was explored. The results are shown in Figure 6. The error lines of the fermented and distilled grains represent the differences in the concentrations of thiols at different locations. The average concentration of thiols in the initial, heaped, fermented, and distilled grains were 65.48, 65.00, 103.78, and 143.19 μg/kg for methanethiol, respectively; 1.15, 1.15, 1.17, and 1.21 μg/kg ethanethiol, respectively; 1.39, 1.53, 1.13, and 1.10 μg/kg for 2-mercapto-1-ethanol, respectively; 0.58, 0.59, 1.40, and 1.97 μg/kg for 2-furfurylthiol, respectively; 1.17, 0.95, 4.07, and 6.35 μg/kg for 2-methyl-3-furanthiol, respectively (Appendix A). While the concentration of ethanethiol did not vary significantly, that of 2-mercapto-1-ethanol decreased during the production process. The concentrations of the other three thiols (methanethiol, 2-furfurylthiol, and 2-methyl-3-furanthiol) increased during fermentation and distillation. The thermal reaction was found to be of great significance for the formation of several thiols. This also explained why, after the fifth round of distillation, there was still a relatively great number of thiols in the fermented grains. The accumulation of precursor substances during the fermentation and accumulation of residual sulfides between rounds led to this phenomenon. Volatile thiol compounds appeared at locations during the entire distillation process of baijiu: one was extracted into the round of baijiu through ethanol extraction, the other was a large amount of residue in the fermented grains, and the third was in other environments. Thus, the fermentation and distillation processes, where the increase in volatile sulfides is higher, are important stages for their widespread production. The stacking process was not the primary stage in the production of these three thiol compounds.

Both the fermentation and distillation processes contributed to the formation of the three thiols to some extent. Therefore, it was necessary to focus on the possible pathways of thiols during both processes. Their presence in alcoholic beverages may be due to enzymatic and non-enzymatic reactions. Enzymatic reactions involve the degradation of sulfur-containing amino acids, formation of fermentation products, and metabolism of some sulfur-containing compounds. Non-enzymatic reactions involve thermochemistry and other chemical reactions of sulfur compounds during brewing and storage. Different pathways for the formation of volatile thiol compounds might occur in parallel during baijiu processing. Cystine, cysteine [26], methionine [27,28], glutathione [29], and thiamine [30] are sources of various sulfur compounds. Their degradations occur via enzymatic or non-enzymatic pathways, and their decompositions cause the formation of additional sulfur compounds [31].

The addition of volatile sulfur compounds to the yeast’s early fermentation stage revealed the complexity of biological and chemical pathways involved in the formation of methanethiol [32]. Methionine [33] and cysteine [34] are the precursors of methanethiol. The Maillard reaction between amino acids and reducing sugars played a key role in the formation pathways of 2-furfurylthiol (FFT) and 2-methyl-3-furanthiol (2M3F). In the Maillard reaction, 4-hydroxy-5-methyl-3(2H)-furanone (NF) and 2-furfural can be generated from ribose using Amadori compounds [35]. 2-Furfural was also one of the main products produced by the reaction of pentose in the presence of amino acids. Hydrogen sulfide (H_2_S) is a Strecker degradation product of cysteine. 2-Furfural and NF are important precursors for the formation of FFT and 2M3F, respectively. In the presence of H_2_S, the hexose degradation products, 2-oxopropanal and hydroxyacetaldehyde, can lead to the formation of FFT and 2M3F. Possible pathway formations in the fermented grains are shown in Figure 7a. 2-Oxopropanal reacts with H_2_S to form mercapto-2-propanone. Depending on the side of the nucleophilic attack, the reaction of mercapto-2-propanone with hydroxyacetaldehyde may lead to the formation of 2M3F or FFT. Mercapto-2-propanone reacts with hydroxyacetaldehyde to form 4,5-dihydroxy-3-mercapto-2-pentanone intermediate product, which is cyclized and eliminated by two water molecules to form 2M3F. In contrast, the methyl nucleophile of mercapto-2-propanone attacks hydroxyacetaldehyde, forming 4,5-dihydroxy-1-mercapto-2-pentanone, which may produce FFT after the cyclization and elimination of water [36]. Possible pathway formations in the fermented grains are shown in Figure 7b. In the case of other sulfur precursor deficiencies, thiamine can act as a sulfur precursor to form 2M3F. Thiamine reacts to form the intermediate 5-hydroxy-3-mercaptopentan-2-one, which undergoes cyclization, dehydration, and oxidation to form 2M3F [37,38]. Detailed information on its formation pathway is presented in Figure 7c. L-cysteine reacts with pentose (ribose, xylose, or arabinose) [39,40] and hexose (glucose) [31] to produce 2-furfurylthiol, 2-methyl-3-furfurylthiol, and other compounds. A variety of bacterial activities, producing β-lyase enzymatic activity, have been shown to degrade cysteine-furfural conjugates to generate 2-furylthiol [26]. The possible pathway formations in the fermented grains are shown in Figure 7d.

Fermented grains contain large amounts of sugars, amino acids, and thiamine; therefore, volatile thiols may be produced during the production of baijiu through the various reaction pathways (Figure 7). The improved method enabled an effective and accurate determination of thiols in fermented grains, which was important for studying the formation pathways of thiols produced during solid-state fermentation, thereby facilitating the division of quality control of baijiu production into different important stages.

## 4. Conclusions

The efficient extraction of thiol compounds during sample preparation was demonstrated using QuEChERS and citrate buffer. Liquid chromatography–tandem mass spectrometry and DTDP derivatization procedures were combined to identify thiols in the fermented grains. Thiols could be derivatized in the MRM mode with good sensitivity and reproducibility using UPLC-MS/MS.

This study found that the concentrations of methanethiol, ethanethiol, 2-furfurylthiol, and 2-methyl-3-furanethiol in fermented grains of sauce-aroma baijiu increased as the number of fermentation rounds increased. Therefore, further analysis will focus on later fermentation rounds for the study of thiol regulation. Volatile thiols had high concentrations in fermented grains after fermentation and distillation. Both fermentation and distillation processes promoted the formation of methanethiol, 2-furfurylthiol, and 2-methyl-3-furanethiol. The fermentation stage may contribute to its precursor substances, which form volatile thiols again during the distillation stage. Thus, more attention can be paid to distillation than just the fermentation process for the regulation of these thiols.

This study was advantageous in analyzing the content of these thiol compounds in baijiu liquor by determining their levels during fermentation and distillation to control the risk of off-odors. This will aid in the real-time monitoring of the alcoholic beverage production process and improve its quality.

## Figures and Tables

**Figure 1 foods-12-02658-f001:**
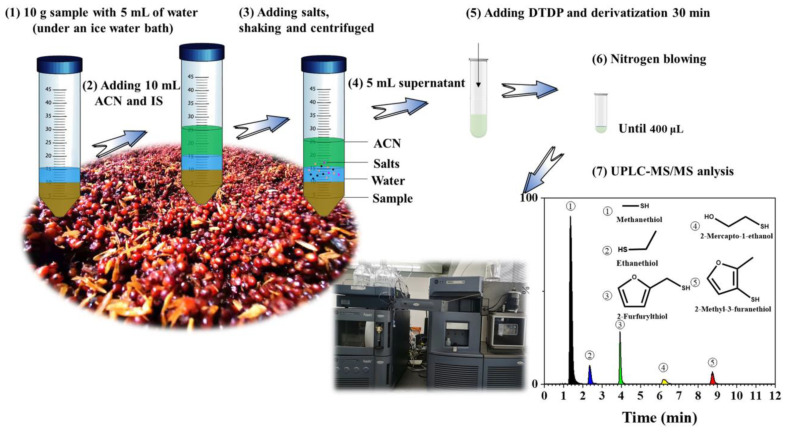
The entire process of fermented grains of baijiu from the pre-treatment to injection analysis.

**Figure 2 foods-12-02658-f002:**
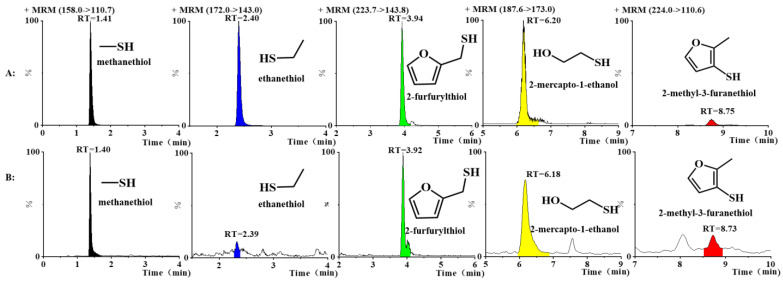
After QuEChERS pre-treatment, MRM chromatograms of five volatile thiols in added and unadded samples. (**A**) Added sample; (**B**) unadded sample.

**Figure 3 foods-12-02658-f003:**
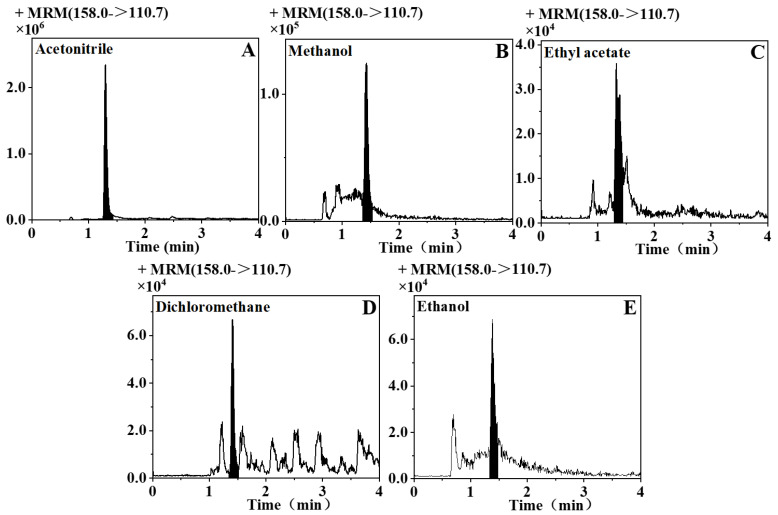
Five common organic extractants were used to extract methanethiol in fermented grains of baijiu, and the final peak pattern was determined using MRM mode. (**A**) Acetonitrile pre-treatment; (**B**) methanol pre-treatment; (**C**) ethyl acetate pre-treatment; (**D**) dichloromethane pre-treatment; and (**E**) ethanol pre-treatment.

**Figure 4 foods-12-02658-f004:**
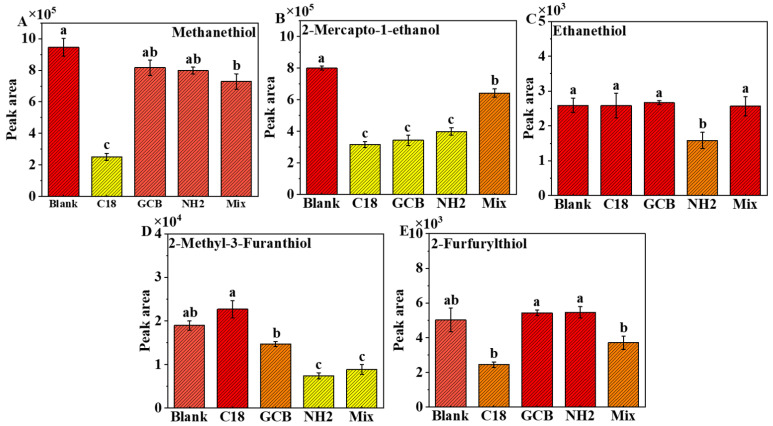
Purification procedures were optimized individually and in combination with four commonly used adsorbents, while the control group did not utilize extractants. (**A**) is the abundance of methanethiol detected under individual and combined purifiers; (**B**) is the abundance of 2-mercapto-1-ethanol detected under individual and combined purifiers; (**C**) is the abundance of ethanethiol detected under individual and combined purifiers; (**D**) is the abundance of 2-methyl-3-furanethiol detected under individual and combined purifiers; (**E**) is the abundance of 2-furfurylthiol detected under individual and combined purifiers. Lowercase letters indicate a significant difference, according to Tukey’s honestly significant difference (*p* < 0.05).

**Figure 5 foods-12-02658-f005:**
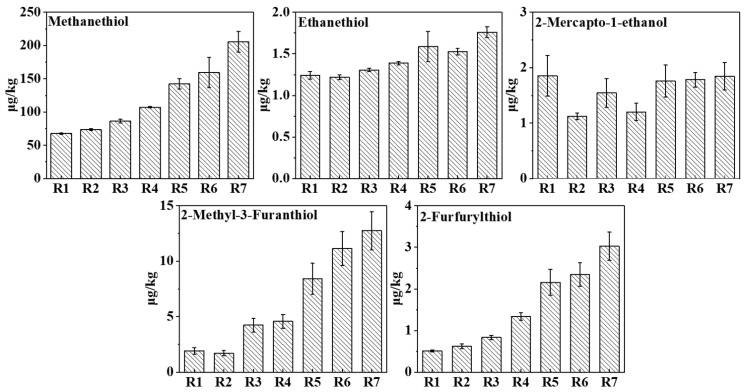
The concentration of thiols tested in fermented grains of baijiu from the first to seventh rounds.

**Figure 6 foods-12-02658-f006:**
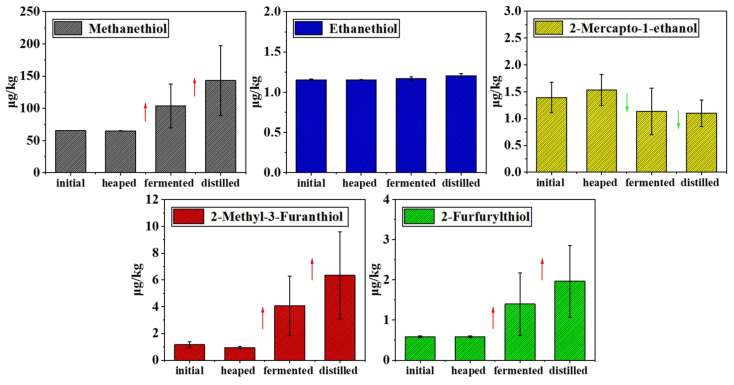
The content of five thiols was tested in fermented grains during the whole fifth round. Red arrows indicate an upward trend, and green arrows indicate a downward trend.

**Figure 7 foods-12-02658-f007:**
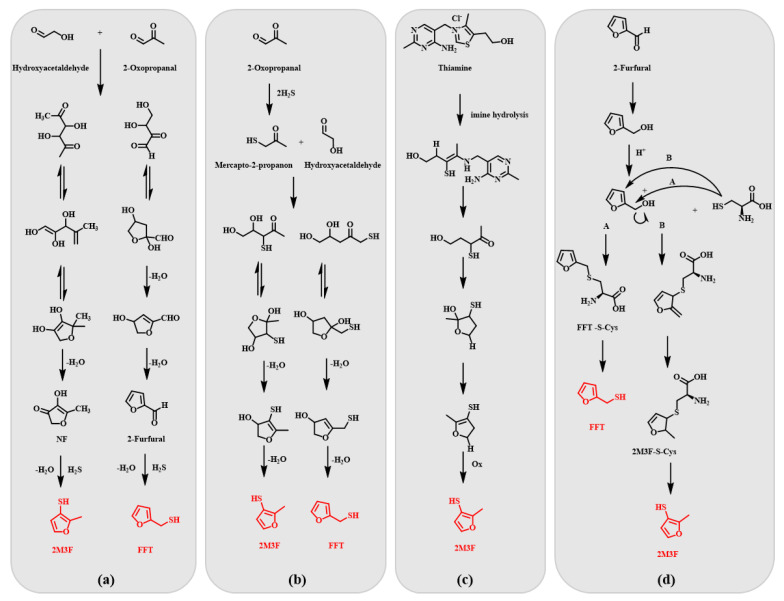
Possible reaction pathways during the fermentation of grains to produce baijiu. (**a**) Pathway for the formation of 2-methyl-3-furanethiol and 2-furfurylthiol with 2-oxopropanal and hydroxyacetaldehyde as precursors; (**b**) Pathway for the formation of 2-methyl-3-furanethiol and 2-furfurylthiol with 2-oxopropanal as a precursor; (**c**) Pathway for the formation of 2-methyl-3-furanethiol with thiamine as a precursor; (**d**) Biological enzymatic reaction pathways for the formation of 2-methyl-3-furanethiol and 2-furfurylthiol.

**Table 1 foods-12-02658-t001:** Linear range, coefficient of determination (R^2^), limit of detection (LOD), and limit of quantification (LOQ) of the derivatized thiols method.

No.	Compounds	Slope	Intercept	Linear Range (μg/kg)	R^2^	LOD (μg/kg)	LOQ (μg/kg)
1	Methanethiol	2.5499	0.9861	10.15–1299.77	0.9907	0.26	0.86
2	Ethanethiol	0.6538	0.0186	0.27–34.24	0.9916	0.017	0.058
3	2-Mercapto-1-ethanol	0.4181	0.00007	0.06–32.42	0.9932	0.004	0.015
4	2-Furfurylthiol	3.8213	0.0079	0.45–28.79	0.9935	0.034	0.11
5	2-Methyl-3-furanthiol	3.272	0.0016	0.17–21.34	0.9900	0.019	0.063

**Table 2 foods-12-02658-t002:** Recovery and precision of the standard approach for derivatized thiols.

Compounds	Spiked Level (μg/L)	Recovery (%)	Precision (RSD, %)	
			Intra-Day	Inter-Day
Methanethiol	5	74.33	0.63	4.28
	50	81.81	2.88	7.24
	500	75.26	3.01	1.96
Ethanethiol	0.3	71.72	4.66	3.43
	3	78.04	1.52	3.89
	30	104.72	1.40	2.85
2-Mercapto-1-ethanol	0.3	85.17	2.02	3.67
	3	74.53	6.04	9.44
	30	72.40	7.72	6.67
2-Furfurylthiol	0.2	82.86	4.51	6.22
	2	100.45	1.42	6.62
	20	102.73	5.60	3.31
2-Methyl-3-furanthiol	0.1	81.12	5.07	4.76
	1	72.55	4.23	3.49
	10	77.79	2.48	8.96

## Data Availability

The data used to support the findings of this study can be made available by the corresponding author upon request.

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
