# Peer review of "Quantification and Distribution of Thiols in Fermented Grains of Sauce-Aroma Baijiu Production Process"

_foods, 2023, doi:10.3390/foods12142658_

Round 1
Reviewer 1 Report (New Reviewer)
The manuscript titled “Quantification and distribution of thiols in fermented grains of sauce-aroma baijiu production process-foods-2426814” deals the develop a method for determining volatile thiol compounds in fermented grains. The subject is interesting and presents some novelty. However, there are some weak points in the research, concerning overall design of the experiment and analytical chemistry. The article needs some revisions.
My remarks about the text are as follows:
Page 2 Line 85: (EtOH, ﹥99.8%) was cghanged to (EtOH ﹥99.8%)
Page 2 Line 98: Add the collecting year of the samples.
Page 3 Line 114: Why authors used a single internal standard (2-phenylethanethiol). Is it enough only one internal standard for all compounds? Please explain.
Page 3 Line 123: Add a reference for sample preparation.
Page 4 Line 147: Only one internal standard is enough for all peak area calculations. Please explain.
Page 4 Line 173: can’t was changed to can not
Please check the reference section for journal requirements.
Minor editing of English language required.
Author Response
Please see the attachment

Reviewer 2 Report (New Reviewer)
Dear Authors,
In general, the manuscript is good to read, the structure of the work is clear and has a sufficient literature review. The first conclusion after reading the work that came to my mind is that the work is a methodical work, like a measurement instruction.
The second conclusion is that there is a lack of statistical analyses, for e.g. PCA analysis.
Some comments below.
- 2.2 Fermented Baijiu Grains - description of the fermentation process is missing
- Lines 167 - 175 these sentences fit more to the methodology than to the description of the results.
- Lines 191 - 194 here is a similar situation, these sentences also fit the methodology and not the description of the results.
- In fact, an interesting description of the measurement results begins only from subsection: "3.4 Measurement of Fermented Grains of Baijiu". Please comment.
- Conclusions section is very short. Some important conclusions from the research should be added. And the authors should reword the Conclusions. It's hard to read in this form.
Author Response
Please see the attachment.

This manuscript is a resubmission of an earlier submission. The following is a list of the peer review reports and author responses from that submission.
Round 1
Reviewer 1 Report
This article dressed the determination of five thiols in wine by LC-MS/MS. QuEChERS was applied in the sample preparation. Validation data was provides for quantitative results. There are some places need to be improved.
Introduction
1. Line 30, suggested to say “These thiols with low thresholds exhibit powerful and characteristic odor.”
2. The English need to be carefully reviewed in order to be considered for publishing. Besides, the current English makes reviewer hard to understand the author’s thinking in the manuscript. I suggest author rewrite the manuscript and make changes, then submit again.
3. Line 110, normally a “-“ is used to describe a 50-mL tube.
4. In 2.4, is capacity centrifuge tube and capacity tube the same? Please make vocabulary consistent.
5. In line 148, normally parent ion/daughter ion or procurer ion/product ion was used.
6. This article similar to “Simple Quantitative Determination of Potent Thiols at Ultratrace Levels in Wine by Derivatization and High-Performance Liquid Chromatography–Tandem Mass Spectrometry (HPLC-MS/MS) Analysis. Anal. Chem. 2015, 87, 2, 1226–1231.” And this should be included in the introduction and the differences may be compared and addressed.
7. In line 212. For statistic, correlation coefficient is R, not R square. R square may be called coefficient of determination.
8. Fig 5. Line 256, the odor thresholds for the five thiols were suggested to be mentioned, and related to the concentration of thiols determined in the wines.
Reviewer 2 Report
Review on “Determination of thiols in fermented grains of Chinese liquor (baijiu) by QuEChERS and UPLC-MS/MS”
In general, the article and the research presented are well written and conducted.
1. Line 15: Substitute “cheap” with “inexpensive”
2. In keywords add “baijiu”
3. In addition to the analysis in lines 49-56, Fluorescence has been recently used for baijiu (Burns, R. L., Alexander, R., et al. (2021). A fast, straightforward and inexpensive method for the authentication of baijiu Spirit samples by fluorescence spectroscopy. Beverages, 7(3), 65.) I suggest to include this.
4. Figure 1 has a picture of MS in very low quality (resolution). Please improve.
5. Figure 2 are also of low resolution. The letters can be hardly read. Please improve.
Reviewer 3 Report
This manuscript is unsuitable for publication as a new method for determining thiols due to several limitations.
It is claimed that it is a new method for determining volatile compounds (line 70).
Are these thiols not present in any other product?
Grains of baijiu from sauce-aroma baijiu-producing regions were collected (line 96), but the region is not specified.
It also needs to be specified how many different locations samples were collected. Surely, you need to collect samples from different locations; even for validation purposes, you must use other grains/products to quantify the following thiols.
Replicates were collected from only one batch. (Line 99)
No optimization of the extraction method.
MRM method is not fully described, and each compound's formula, mass, and product ions need to be included (Figure 2 is unsuitable in this scenario). Whenever possible, provide PubChem IDs.
When developing a method, you also need to optimize LC conditions.
No supporting information is provided.